# Indigenous Lactic Acid Bacteria Isolated from Raw Graviera Cheese and Evaluation of Their Most Important Technological Properties

**DOI:** 10.3390/foods12020370

**Published:** 2023-01-12

**Authors:** Evdoxios Psomas, Ioannis Sakaridis, Evridiki Boukouvala, Maria-Anastasia Karatzia, Loukia V. Ekateriniadou, Georgios Samouris

**Affiliations:** 1Department of Hygiene and Technology of Food of Animal Origin, Veterinary Research Institute, Hellenic Agricultural Organization-Demeter, Campus of Thermi, 57001 Thessaloniki, Greece; 2Research Institute of Animal Science, Hellenic Agricultural Organization-Demeter, Paralimni, 58100 Giannitsa, Greece

**Keywords:** lactic acid bacteria, graviera cheese, starter cultures, raw-milk cheese

## Abstract

The aim of the present study was to characterize LAB isolates from raw-milk cheeses, to evaluate some of their technological properties and to select a few ‘wild’ LAB strains that could potentially be used as starter cultures. LAB strains were isolated and identified from raw milk, curd, and cheese at 30, 60, and 90 days of ripening. A total of 100 strains were isolated, 20 from each phase of ripening. All isolates were tested for acidification ability, curd formation, and aroma production at 32 °C and 42 °C after 24 and 48 h. Following the acidification test, 42 strains were selected for identification and characterization of their technological properties. A high proportion of lactic acid bacteria and Gram + cocci were found throughout the cheese-making process. Enterococci reached their maximum proportion on the 7th day of ripening while Lactobacilli increased significantly during the first month of ripening. Forty-two strains were identified by phenotypic, biochemical, and molecular techniques. Lactococci were predominant in raw milk and curd while Lactobacilli in the ripening of the cheese. Four LAB strains including one *Leuconostoc pseudomenteroides*, two *Lacticaseibacillus paracasei* subsp. *paracasei* and one *Enterococcus hirae*, were proposed for their potential use as starters or secondary cultures.

## 1. Introduction

Cheese is a rich source of essential nutrients and specifically contains proteins of high biological value, peptides, amino acids, fat, fatty acids, vitamins, and minerals. In most cases, cheese producers subject raw milk to a thermal treatment equal to pasteurization in order to control the pathogenic bacteria that are present in it and to improve its safety for consumption. However, there are some cheese producers that prefer the traditional method of cheese making and use unpasteurized milk for their dairy products. For these cheesemakers and for their consumers, traditional manufacturing of cheese from raw milk is an important feature of cheese making culture. Industrially produced cheeses are known to lack traditional flavors, or at least some characteristic flavors, due to pasteurization of the milk and the use of commercial starter cultures [1,2]. On the contrary, cheeses made with raw milk reveal different flavors and texture characteristics to the matured cheese, exhibiting greater overall flavor intensity and broader flavor profiles. Larger amounts of aromatic compounds such as acids, aldehydes, alcohols, esters, and sulfur compounds are produced during the manufacturing and ripening of raw-milk cheeses compared to the cheeses made from pasteurized milk [3]. The diversity of species and strains of local and specific indigenous milk microflora is the main factor that affects the unique sensory characteristics of raw-milk cheeses [4,5].

Lactic acid bacteria (LAB) have been used for a long time to preserve, improve the quality, and modify the flavor of dairy products. These include bacteria from the genera *Lactobacillus*, *Lactococcus*, *Leuconostoc*, *Pediococcus*, and *Streptococcus*, which are the main agents responsible for milk fermentation and cheese production [6]. The microflora of cheese is comprised of two major groups: starter lactic acid bacteria and secondary microorganisms. The starter lactic acid bacteria are responsible for the acid production during cheese making and have also an important role during the ripening process [7]. The secondary microorganisms usually come from the autochthonous milk or environmental microbiota and generally play an important role during ripening by enhancing flavor development or sometimes deteriorating cheese quality [8]. They consist of non-starter lactic acid bacteria (NSLAB) that are present in most cheese varieties and other bacteria, yeasts, and/or molds, which grow internally or externally and are usually found only in specific cheeses [7].

Over the last few years, there is a particular interest in the microbiota of raw-milk cheeses and generally traditional cheeses for new LAB strains that could potentially complement or even replace the strains that are currently used in commercial starter cultures [5,9]. These traditional cheeses contain LAB strains of a vast phenotypic and genetic microbial diversity that could potentially have multiple biotechnological applications [5,10,11]. Such LAB strains have been isolated, characterized, and used as starters and adjunct cultures for cheese production in several studies [12,13,14,15].

The aim of the present study was to isolate, identify, and characterize LAB isolates from raw-milk starter-free cheeses with typical sensorial profiles, to evaluate some of their technological properties with those of commercial cultures and, finally, to select a few ‘wild’ LAB strains that could potentially be used as starter cultures.

## 2. Materials and Methods

### 2.1. Cheese Making

Raw cow’s milk was warmed at 35 °C and the quantity of rennet (Bioren, Italia) required to curdle the milk within 30–35 min was added. No defined starter cultures were added to this batch (batch-R). At the same time another batch was produced with heat treated cow’s milk (batch-P). Raw cow’s milk was pasteurized (65 °C, 30 min) and then cooled to 35 °C, and a starter culture (mesophilic and thermophilic, MALP, Italy), and the same rennet as with the previous batch was added. Approx. 35–40 min after the addition of the rennet, the milk of both batches (R and P) was coagulated. The curd was then cut into small pieces (1 cm × 1 cm × 1 cm in size) and cooked for 20–30 min at 50–52 °C with continuous stirring. It was then placed in molds and pressed for 1–2 h (1–2 times their weight) before being inverted for molding and drainage of the cheese. After 24 h and when the pH of the cheese was 5.4, it was transferred into brine (18–20 Be) at a temperature of 13–14 °C. For each kilo of cheese, 8–9 h of incubation in the above brine were needed. The cheeses were transferred to the ripening area (for 10–15 days at 13–14 °C), with inversions every 1–2 days. After the end of the 1st ripening period, the cheeses were air-sealed closed and kept at a lower temperature (6–8 °C) for at least 3 months to continue ripening (2nd ripening period). For each batch, samples of milk and cheese were taken after 2, 7, 15, 30, 60, and 90 days. The cheese making trials were carried out five times in total and for each trial, microbiological and chemical analysis of the milk and cheese were performed.

### 2.2. Samples

Analyses were carried out for the raw milk and curd on the day of the cheese making and for cheese on the 30th, 60th, and 90th day of ripening. Duplicate samples were collected.

### 2.3. Isolation of Bacteria and Growth Conditions

Duplicate milk (10 mL), curd, and cheese samples (10 g) were homogenized with 90 mL of Buffer Peptone Water (BPW) (Oxoid, Basingstoke, UK) and diluted, using a Stomacher Lab-Blender 400 (Seward) for approximately 5 min at 180 rpm. Then, aliquots were plated and incubated as follows: presumptive mesophilic and thermophilic lactic acid bacteria on MRS medium (MRS) (Oxoid, Basingstoke, UK) at 30 °C for 24–48 h, under aerobic conditions; presumptive mesophilic and thermophilic cocci on M17 medium (M17) (Oxoid, Basingstoke, UK) at 30 °C for 24–48 h, under aerobic conditions; and presumptive lactobacilli on Rogosa medium (Oxoid, Basingstoke, UK) at 30 °C for 5 days, under aerobic conditions; and enterococci on kanamycin-esculin-azide (KAA) (Oxoid, Basingstoke, UK) medium at 37 °C for 24–48 h. Each plate was macroscopically examined and then five colonies with optical differences were randomly collected. A total of 100 colonies (20 per stage during cheese making) were selected from the plates and then were purified by streaking in their corresponding isolation medium. Before their storage, all (100) isolates were preliminarily tested microscopically for their Gram staining, catalase activity, and spore staining. Gram-positive, catalase-negative, non-spore-forming bacteria were stored at −18 °C and −80 °C in duplicates, in polypropylene tubes (1.5 mL) containing their corresponding isolation medium with added glycerol (70:30).

### 2.4. Technological Characterization

The isolated strains were studied for their technological reference according to methods described by several researchers [16,17,18,19].

For the determination of the *proteolytic activity* of the strains, after being cultured in MRS broth, they were streaked on plates with PCA (Oxoid, Basingstoke, UK) + 10% skim milk powder. Incubation at 37 °C for 24–48 h and 7 °C for 10 days followed. In a positive test, a clear zone was formed around the colonies [17,18,19].

The acidifying activity of isolates from milk was tested by inoculating 50 mL UHT milk with an activated culture 1% (*v*/*v*). The pH was measured by a pH meter (Consort C5010, Belgium) at 0, 24, and 48 h after incubation at 32 °C and 42 °C [16,17]. At the same time, additional characteristics were observed, such as curd formation, serum separation, and aroma formation [19].

The enzymatic hydrolysis of tributyrin In butyric acid and glycerol (*lipolysis*) was tested using the Rosco Tributyrin Diatabs TM diagnostic tablets (Rosco, Taastrup, Denmark) according to the manufacturer’s instructions.

### 2.5. Phenotypic Characterization of Isolates

Both cocci and rod-shaped strains were phenotypically characterized according to the following criteria: appearance by Gram staining; ability to grow after incubation in MRS broth (Oxoid, Basingstoke, UK) at 10 °C and 45 °C for cocci and 15 °C and 45 °C for rods; production of CO_2_ from glucose using inverted Durham tube in MRS broth and incubation at 30 °C. For cocci, distinguished extra tests were specifically used: tolerance at 6.5% NaCl, ability to grow in MRS substrate with vancomycin, and esculin hydrolysis [20,21]. The ability to ferment carbohydrates was studied by observing the gas production and the color changes of the tubes that contained the sugar substratum for fermentation (37 °C for 24 h) (Sigma-Aldrich, St. Louis, MO, USA).

### 2.6. Molecular Identification

#### 2.6.1. DNA Extraction

Bacterial cells were grown overnight in 10 mL of their corresponding isolation medium, and DNA was extracted according to the manufacturer’s instructions for Gram positive bacteria (Pure Link^TM^ Genomic DNA Extraction Kit, Life Sciences-Thermo Fisher Scientific, Waltham, MA, USA).

#### 2.6.2. Multiplex PCR

Isolates of presumptive Lactococci, Lactobacilli, Streptococci, and Leuconostocs were assigned to the genus level by Multiplex PCR as described by Tsirigoti et al. [22].

Identification of presumptive Enterococci at the genus and species levels were also accomplished by multiplex PCR using four pairs of primers for the detection of *E. faecalis*, *E. faecium*, *E. hirae*, and *E. durans* species as described by Jackson et al., 2004 [23] and a pair of primers for the Enterococcus genus according to Deasy et al. [24].

Multiplex PCR reactions were performed in a final volume of 10 μL, containing 1 × KAPA 2G Multiplex PCR Mix (KAPA Biosystems), 300 nM of each primer, and 80–100 ng DNA. The protocol included an initial denaturation step for 2 min at 95 °C, followed by 30 cycles of three steps: denaturation at 95 °C for 30 s, annealing of the primers at 60 °C for 30 s, and elongation at 72 °C for 30 s. The final elongation step occurred at 72 °C for 7 min.

#### 2.6.3. Sequencing

The 16S rRNA gene was amplified by PCR with the primers LABSeqF: GCT CAG GAY GAA CGC YGG and LABSeqR: CAC CGC TAC ACA TUR ADT TC [25]. The protocol included an initial denaturation step for 3 min at 95° C, followed by 35 cycles of three steps: denaturation at 98 °C for 20 s, annealing of the primers at 58 °C for 25 s, and elongation at 72 °C for 45 s. The final elongation step occurred at 72 °C for 5 min. The PCR products were purified using the Gel Extraction Kit (PureLink^TM^ Quick Gel Extraction kit, Invitrogen, Thermo Fisher Scientific, Waltham, MA, USA) and sequenced using the same primers in a ABI3500 genetic analyzer. Each strain was identified by comparing DNA sequence obtained with those in the NCBI database

## 3. Results

### 3.1. Technological Characterization

In total, 100 colonies of LAB were selected from the five cheese making trials: 20 from each cheese-making stage and 5 from each medium (MRS, M17, Rogosa, KAA) within each stage. The selection was based on the visual differentiation (color, size, shape) between the isolated colonies. The above colonies were initially assigned to a genus level by morphology and simple physiological tests following the criteria of Sharpe [26] using morphological, phenotypic, and biochemical methods [27]. All strains were observed morphologically under the microscope after Gram staining and classified into Gram + cocci (52%) and bacilli (48%). This was followed by the catalase test and spore staining where all strains that were characterized as non-sporogenic, catalase negative were stored in the appropriate nutrient medium with glycerol as mentioned in the materials and methods.

These isolated strains were tested for their ability to produce acid in milk at 32 °C and 42 °C. At the same time, clot formation, aroma, and excretion of serum during clotting were observed [17,28]. The aroma was appreciated by a group of six panelists [19]. Based on the desired technological characteristics, 42 strains were selected for identification and further study. Of the 42 strains, 7 came from raw milk, 9 from curd after salting, 12 from 30-day-ripened cheese, 8 from 60-day-ripened cheese, and 6 from 90-day-ripened cheese (Table 1).

Only two of the selected strains (KAA-2 and KAA-3) belonging to the species *Enterococcus faecalis* and *Enterococcus gallinarum* showed proteolytic activity at 37 °C, but not at 7 °C. At 7 °C, 8 strains were found to be positive, of which 2 were isolated from raw milk, 2 from curd after salting, and the rest from cheese during the ripening stage (Data not shown).

In terms of their lipolytic activity, 25 strains were positive in the tributyrin test, i.e., they had the ability to break down tributyrin into butyric acid and glycerol. Most test-positive strains were isolated from cheese during the ripening stage.

### 3.2. Phenotypic Identification

As far as lactococci were concerned, for the purpose of their identification, they were divided into two groups, the Gram+, catalase negative cocci and the cocci isolated from the selective medium Kanamycin Esculin Azide Agar (KAA agar). All isolates were tested for growth at both temperatures (10 °C, 45 °C), 6.5% NaCl, and vancomycin-modified MRS medium. Gas production from glucose was tested and the ability to hydrolyze esculin was examined. When examined under the microscope, no strain was observed to form tetrads, a morphology characteristic of the genera *Tetragenococcus* and *Pediococcus* [29]. Isolates from the KAA agar medium showed a positive growth test at 10 °C and 45 °C. All isolates were able to grow in the presence of 6.5% NaCl and hydrolyze esculin, while not producing gas from glucose. According to Schillinger & Lücke [29], these strains belong to the genus *Enterococcus*. For this specific genus, the sugars examined were: melibiose, raffinose, rhamnose, mannitol, sorbitol, sucrose, and dulcitol [30,31]. Based on the results shown in Table 2, isolates 1, 2, 4, 5, and 8 were identified as *Enterococcus faecalis*, with the ability to ferment the sugars sucrose and melibiose being strain-dependent. Finally, strains 3 and 6, were identified as *Enterococcus gallinarum* and strains 7 and identified as *Enterococcus faecium*.

Two strains (RO32.1, RO32.2) produced gas from glucose, showed resistance to vancomycin, and were able to hydrolyze esculin. They also grew at 10 °C but not at 45 °C. These strains belong to the genus *Leuconostoc* [32,33]. The sugars used to identify these strains were: maltose, galactose, raffinose, arabinose, trehalose, and fructose [34,35]. Both strains were identified as *Leuconostoc pseudomesenteroides*.

Finally, six strains with a positive test at 10 °C, negative at 45 °C, inability to grow at a concentration of 6.5% NaCl, and gas production from glucose while being able to hydrolyze esculin were assigned to the genus *Lactococcus*. Three strains (MRS-1, MRS-2, M17–1) were positive for the fermentation of maltose, ribose, galactose, lactose, and salicin, and negative for melibiose and melezitose and were assigned to the genus *Lactococcus garviae* [36]. Another three strains (MRS-4, M17-3, M17-1) with a positive test for the sugars maltose, ribose, galactose, and lactose, and negative for melibiose, melezitose, and salicin were assigned to the genus *Lactococcus lactis* subsp. *lactis*.

Strains M17-2 and M17-7 showed the same profile of phenotypic characteristics, i.e., negative test at 10 °C and in 6.5% NaCl concentration, esculin hydrolysis, and positive at 45 °C. The sugar fermentation profile was positive for the sugars maltose, salicin, melibiose, galactose, fructose, raffinose, and sucrose, and negative for galacticol, rhamnose, mannose, melezitose, and sorbitol. These strains were identified as from the genus *Streptococcus* spp.

As far as Bacilli were concerned, they were tested for gas production from glucose and their ability to grow at 15 °C and 45 °C. All strains showed growth at 15 °C, while at 45 °C ten strains grew (MRS-3, MRS-6, MRS-7, MRS-10, MRS-12, MRS-13, RO-4, RO-6, RO-8, RO-9). Gas from the breakdown of glucose was produced by strains MRS-8, MRS-9, and M17-6. The sugars examined for all strains were the following: rhamnose, lactose, xylose, trehalose, ribose, and cellobiose. The sugars used to identify the strains that did not produce gas were mannitol, turanose, and raffinose, while the additional sugars used for the glucose-fermenting strains were mannose, cellobiose, amygdalin, and fructose.

Strains with a positive test in the fermentation of the sugars lactose, mannitol, turanose, trehalose, ribose, and rhamnose, and negative for xylose, melibiose, and raffinose were identified as *Lacticaseibacillus rhamnosus* [37]. The strains that showed a similar profile to the above with the only difference being a negative test in rhamnose fermentation were identified as *Lacticaseibacillus paracasei* subsp. paracasei [38]. Of the isolates, two that tested positive for gas production from glucose (MRS-8, MRS-9) and produced acid from lactose, xylose, and fructose but did not produce acid from rhamnose, trehalose, melibiose, cellobiose, and amygdalin were identified as *Lactobacillus parabrevis* according to Vancanneyt et al. [39], and one strain (M17-6) was identified as *Lactobacillus kefiri*. The analytical results are shown in Table 3.

### 3.3. Molecular Identification

From the electrophoresis of the Multiplex PCR products (Figure 1), it can be seen that all lactobacilli strains showed a band at 270 base pairs, the same size as the standard strain *Lacticaseibacillus paracasei* subsp. *paracasei* that was used. Similarly, the lactococcal strains all gave a characteristic band (386 bp) identical to the standard strains used (Figure 2). The two strains of streptococci showed a double band during electrophoresis of the amplified product; therefore, the strains were not considered pure. Figure 3 shows the strains MRS-8, MRS-9, and M17-6 which, based on biochemical tests, were identified as lactobacilli, while in Multiplex PCR it appeared that the first two belong to the genus Lactococci and the latter to the genus *Leuconostoc*. As for enterococci (Figure 4), all strains showed a band characteristic of the genus enterococci (780 bp) and specifically strains 5, 6, 9, 10, and 13 were identified as *Enterococcus faecalis*; 11 and 12 as *Enterococcus hirae*; and the species of strains 7 and 8 could not be identified. The comparative results of the biochemical and molecular identification of all bacteria are shown in Table 4.

### 3.4. Sequencing

Representative strains for each species were selected to amplify and sequence part of the 16S rRNA gene (650 bp). The sequence of the selected strains was compared with those from the data bank (GenBank) using the BLAST algorithm. The sequencing results are shown in Table 5.

Strain M17-6 gave high homology rates after the sequence search in the database with the species *Lentilactobacillus otakiensis*, *Lentilactobacillus parabuchneri*, and *Lentilactobacillus kefiri*. By comparing the phenotypic, biochemical, and molecular results, we concluded that the strain belongs to the species *Lentilactobacillus kefiri*.

Similarly, strain MRS-10 presented high homology rates with the species *Lacticaseibacillus rhamnosus* and *Lacticaseibacillus paracasei* subsp. paracasei. This strain was identified as *Lacticaseibacillus paracasei* subsp. paracasei based on biochemical tests as it was not able to ferment the rhamnose, a sugar characteristic for the differentiation of *Lacticaseibacillus rhamnosus* with *Lacticaseibacillus paracasei* subsp. *paracasei*.

## 4. Discussion

Cheeses derived from raw milk are characterized by organoleptic characteristics, which are partly due to the diversity of the microbial flora, i.e., to the action of LAB and NSLAB during the ripening of the cheese [40,41]. LAB may have been intentionally added to the cheese as a starter culture or may have entered accidentally from milk to the machinery and the immediate environment of the plant during cheese making (NSLAB). The initially small population of random, non-starter LAB becomes the dominant bacterial population in the ripened cheese. The study of NSLAB isolated from raw milk or its dairy products is important, and the aim is to isolate strains that can possibly be used as starter cultures or additional cultures with the aim of improving the organoleptic characteristics of cheeses [42].

Lactococci were predominant in the raw milk and curd; specifically, 3 strains of *Lactococcus garviae* and 4 strains of *Lactococcus lactis* subsp. *lactis* out of a total of 14 strains. Furthermore, 2 strains of *Leuconostoc pseudomesenteroides* were isolated from raw milk. Picon et al. reported a predominance of lactococci over lactobacilli on the first day of cheese making with a reversal of the ratio on the sixtieth day of cheese making and a predominance of lactobacilli [43]. Similarly, other researchers reported an increase of lactococci from Graviera during the coagulation stage and their reduction during the ripening of the cheese [44]. Strains of *Lc. garviae* were isolated and identified from raw milk but not from coagulated milk, whereas strains of *Lc. lactis* subsp. *lactis* were predominant. This is probably since strains of the species *Lc. garvieae* are more sensitive to extreme conditions such as low temperature and acidity [45]. Furthermore, *Lc. lactis* subsp. *lactis* prevailed in the curd but was not found during the ripening stage and this is probably due to the high salt concentration in the cheese [46]. Therefore, *Lc. lactis* subsp. *lactis* is often found in raw-milk cheeses [47,48].

Lactobacilli prevailed during the ripening of the cheese, a finding that is in agreement with similar studies in graviera cheese [44,49]. The dominant species in terms of lactobacilli was *Lacticaseibacillus paracasei* subsp. *paracasei* (15/23), especially in the final stage of maturation, followed by *Lacticaseibacillus rhamnosus* (5/23), while 2 strains of *Levilactobacillus parabrevis* and 1 *Lactobacillus kefiri* were also isolated out of a total of 23 strains. Mesophilic lactobacilli are an important part of the natural microflora of traditional Greek cheeses [50] and prevail during the ripening stage [51,52]. The effect of lactobacilli (*Lb. helveticus*, *Lb. paracasei* subsp. *Paracasei*, *Lb. delbrueckii* subsp. *lactis*, *Lb. plantarum*, etc.) on the organoleptic characteristics of cheeses when added as cultures (starter or secondary) is very important as they enhance the aroma and flavor of cheeses [53,54,55,56].

In other studies related to the diversity of lactic acid bacteria isolated from graviera cheese made from raw milk, *Lacticaseibacillus paracasei* subsp. *paracasei* was found to prevail at the ripening stage [49,57,58]. Veljovic et al. reported that the predominant NSLAB species in semi-hard brine cheese produced from raw cow’s milk are *Lb. paracasei*, *Lb. brevis*, and *Lb. plantarum* while Sánchez et al. found *Lb. paracasei* subsp. *paracasei* [48,59] in Spanish raw-milk cheese. In Cheddar cheese, the predominant lactic acid microflora was *Lb. paracasei* and *Lb. rhamnosus* [60].

The predominant enterococci species in our study was *Enterococcus faecalis*. Strains of the species *Enterococcus gallinarum* and *Enterococcus hirae* were also isolated. Strains of the species *Enterococcus faecium*, *Enterococcus durans*, and *Enterococcus faecalis* were isolated during the ripening stage in gruyere prepared from heated milk [49]. Our results are also in accordance with the findings of other studies where the predominant species of enterococci in raw-milk cheeses were *E. faecalis*, *E. faecium*, and *E. durans* [44,57,61,62].

Two strains of enterococci were found to have proteolytic activity at 37 °C. At 7 °C, proteolytic activity was shown by the strain RO-2 (*Leuconostoc pseudomesenteroides*), four strains of *Lacticaseibacillus paracasei* subsp. *paracasei*, and the strain MRS41.1 (*Lactococcus lactis* subsp. *lactis*). Strains of the species *Lacticaseibacillus paracasei* subsp. *paracasei* have been found to have proteolytic properties by other researchers [63]. Proteolysis is considered as the key process determining the rate of flavor and texture development in most cheese varieties [49].

According to the results of tributyrin test, strains of lactococci, lactobacilli, and enterococci were found to be capable of breaking down tributyrin. The strains of enterococci showed lipolytic activity which has also been reported by other researchers [64]. The strains during the ripening stage appeared to be more lipolytic. Lactococci did not show any lipolytic ability except for the strains MRS41.1 *Lc. lactis* subsp. *lactis* and RO-2 *Leuconostoc pseudomesenteroides*.

The selection criteria of NSLAB strains for their use as starter cultures are: (1) the ability to produce acid in milk, (2) proteolytic activity and aroma production, and (3) lipolytic activity [51,65]. The main purpose of the use of a strain as adjunct culture is the enhancement of the aroma and flavor of the cheese through its proteolytic and/or lipolytic properties as well as its ability to speed up the ripening stage. Based on the above criteria, four NSLAB strains were proposed for possible use as adjunct cultures.

The strain RO-2 isolated from Rogosa agar and identified as *Leuconostoc pseudomesenteroides* showed a desirable profile of technological properties as it was able to reduce the pH value > 1.5 unit at 32 °C, as well as showing proteolytic and lipolytic activity (tributyrin) so it is recommended for use as a starter culture. The MRS-10 strain identified as *Lacticaseibacillus paracasei* subsp. *paracasei* showed no proteolytic activity and was negative in the tributyrin test but managed to lower the pH by more than 2 units at both temperatures. In addition, it formed a cohesive gel and a pleasant aroma, especially at 32 °C. Therefore, it is recommended to be used as a starter culture. The strain RO-11 belonging to the species *Lacticaseibacillus paracasei* subsp. *paracasei* showed proteolytic and lipolytic activity with a strong cheese aroma. It showed low acidification capacity, therefore this strain is recommended for use as an additional culture or in combination with a strain that has strong acidification capacity as a starter culture. The strain KAA-9 identified as *Enterococcus hirae*, has proteolytic and lipolytic activity, low acidification, and coagulating capacity with a pleasant aroma at both temperatures. Thus, it is proposed to be studied for its possible pathogenicity in order to be used as an adjunct culture. These strains are proposed for further study of their technological properties. We also propose to study the possibility of using them in combination with other bacteria in mixed cultures.

## 5. Conclusions

Microbial communities play an essential role in the control of the sensory qualities of cheese. They are more diverse and complex in raw-milk cheeses for which milk does not undergo any treatment to reduce its microflora. They contribute to the development of a typical cheese taste and flavor. Therefore, it is important for raw-milk cheeses to maintain/have high taxonomic diversity in indigenous cheese communities and diverse cheese making practices. A high diversity of raw-milk microbiota is the key for allowing the cheese to develop its particular characteristics including low pathogen risk and diversification of gustatory characteristics.

Raw-milk cheeses have a variety of indigenous LAB that could potentially be used as starter cultures or secondary cultures. The evaluation of the technological properties of the LAB isolated from a raw-milk graviera cheese, such as acidification ability, proteolytic activity, and lipolytic activity revealed that four LAB strains including one *Leuconostoc pseudomenteroides*, two *Lacticaseibacillus paracasei* subsp. *paracasei*, and one *Enterococcus hirae*, have the potential to be used as starters cultures or in combination with other bacteria in mixed cultures.

## Figures and Tables

**Figure 1 foods-12-00370-f001:**
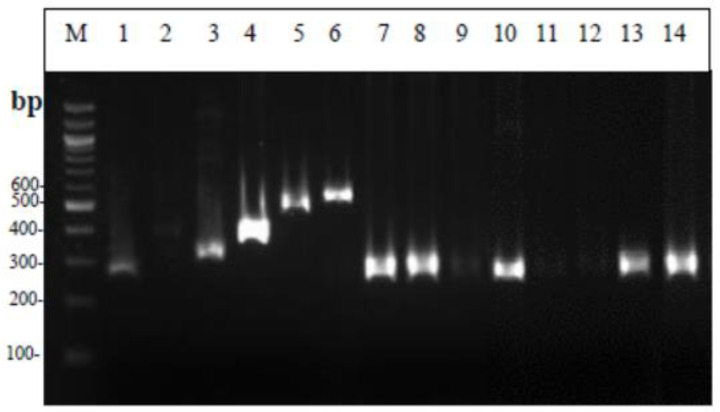
Results from Multiplex PCR. Μ: 100 bp DNA ladder, 1: *Lacticaseibacillus* subsp. *paracasei* ATCC 25302, 2: *Lactococcus garvieae* subsp. *garvieae* DSM 20684, 3: *Lactobacillus pentosus* NCFB 363, 4: *Lactococcus lactis* DSM 0718, 5: *Streptococcus thermophilus* LMG 13565, 6: *Leuconostoc pseudomesenteroides* DSM 20193, 7: MRS-12, 8: MRS-10, 9: MRS-6, 10: RO-8, 11: ΜRS-3, 12: RO-9, 13: MRS-7, 14: RO-11.

**Figure 2 foods-12-00370-f002:**
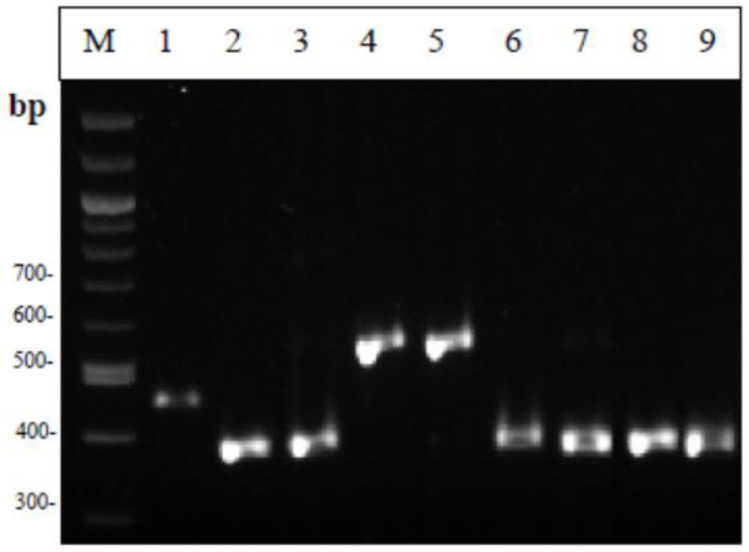
Results from the Multiplex PCR identifying the four genera Lactococcus, Lactobacillus, Streptococcus and Leuconostoc. Μ: 100 bp DNA ladder, 1: *Streptococcus thermophilus* LMG 13565, 2: *Lactococcus lactis* DSM 0718, 3: *Lactococcus garvieae* subsp. *garvieae* DSM 20684, 4: *Leuconostoc pseudomesenteroides* DSM 20193, 5: RO-1, 6: M17-3, 7: MRS-1, 8: MRS-4, 9: MRS-5.

**Figure 3 foods-12-00370-f003:**
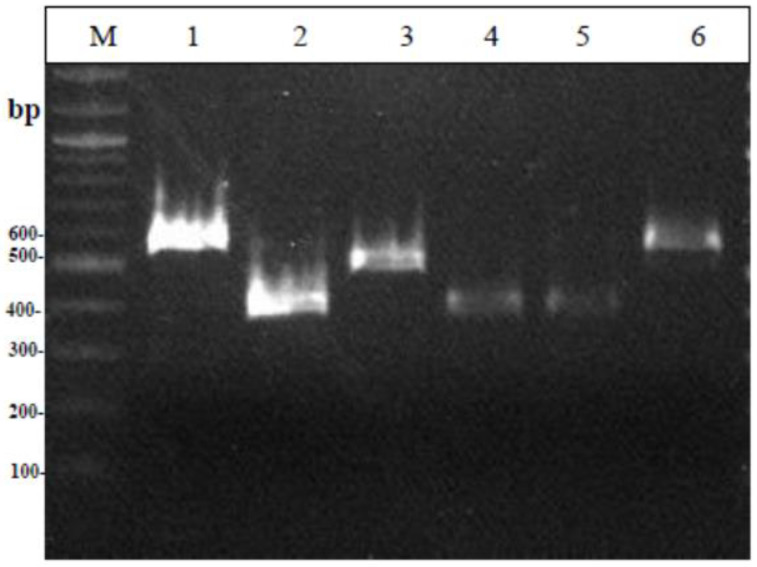
Results from the Multiplex PCR identifying the four genera Lactococcus, Lactobacillus, Streptococcus and LeuconostocΜ: 100 bp DNA ladder, 1: *Leuconostoc pseudomesenteroides* DSM 20193, 2: *Lactococcus garvieae* subsp. *garvieae* DSM 20684, 3: *Streptococcus thermophilus* LMG 13565, 4: MRS-8, 5: MRS-9, 6: M17-6.

**Figure 4 foods-12-00370-f004:**
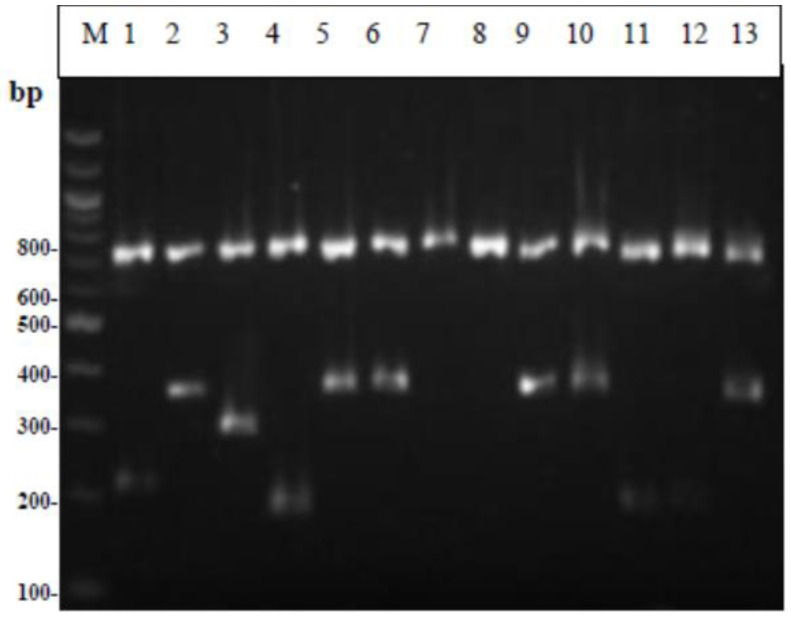
Results from the Multiplex PCR identifying Enterococcus genus and speces. Μ: 100 bp DNA ladder, 1: *Enterococcus faecium* DSM 20477, 2: *Enterococcus faecalis* DSM 20478, 3: *Enterococcus durans* DSM 20633, 4: *Enterococcus hirae* DSM 20160, 5: KAA-1, 6: KAA-2, 7: KAA-3, 8: KAA-6, 9: KAA-4, 10: KAA-5, 11: KAA-7, 12: KAA-8, 13: KAA-9.

**Table 1 foods-12-00370-t001:** Technological properties of LAB strains.

Isolation Stage	Media-Strain	CLOT 32 °C	CLOT 42 °C	AROMA 32 °C	AROMA 42 °C
Raw milk	RO-1	+	−	yes	yes
	RO-2	+	+	yes	no
	M17-1	+	−	yes	yes
	MRS-1	+	+	yes	yes
	MRS-2	+	+	yes	yes
	RO-3	+	+	yes	yes
	KAA-1	+	+	yes	yes
Cheese after salting	MRS-3	+	+	yes	yes
	RO-4	+	+	yes	yes
	RO-5	+	+	yes	yes
	MRS-4	+	+	yes	yes
	MRS-5	+	+	yes	yes
	M17-2	+	+	yes	yes
	M17-3	+	+	yes	yes
	KAA-2	+	+	yes	yes
	KAA-3	+	+	yes	yes
Cheese (30 days)	MRS-6	+	+	yes	yes
	MRS-7	+	+	yes	yes
	RO-6	+	+	yes	yes
	M17-4	+	−	yes	yes
	M17-5	+	−	yes	no
	RO-7	+	+	yes	yes
	MRS-8	+	+	yes	yes
	MRS-9	+	+	yes	yes
	M17-6	+	+	yes	yes
	KAA-4	+	+	yes	yes
	KAA-5	+	+	yes	yes
	KAA-6	+	+	yes	yes
Cheese (60 days)	MRS-10	+	+	yes	yes
	RO-8	+	+	yes	yes
	RO-9	+	+	yes	yes
	MRS-11	+	+	yes	yes
	RO-10	−	+	no	yes
	RO-11	+	+	yes	yes
	M17-7	+	+	yes	yes
	KAA-7	+	-	yes	yes
Cheese (90 days)	MRS-12	+	+	yes	yes
	MRS-13	+	+	yes	yes
	RO-12	+	+	yes	yes
	RO-13	−	−	yes	no
	KAA-8	+	+	yes	yes
	KAA-9	+	−	yes	yes

**Table 2 foods-12-00370-t002:** Identification of isolated cocci using classical techniques.

Growth Media	Strain	Species
MRS	1	*Lc. garvieae*
MRS	2	*Lc. garvieae*
MRS	4	*Lc. lactis* subsp. *lactis*
MRS	5	*Lactococcus* spp.
M17	1	*Lc. garvieae*
M17	4	*Lc. lactis* subsp. *lactis*
M17	2	*Streptococcus* spp.
M17	7	*Streptococcus* spp.
RO	1	*Lu. pseudomesenteroides*
RO	2	*Lu. pseudomesenteroides*
KAA	1	*Ent. faecalis*
KAA	2	*Ent. faecalis*
KAA	3	*Ent. gallinarum*
KAA	4	*Ent. faecalis*
KAA	5	*Ent. faecalis*
KAA	6	*Ent. gallinarum*
KAA	7	*Ent. faecium*
KAA	8	*Ent. faecalis*
KAA	9	*Ent. faecium*

**Table 3 foods-12-00370-t003:** Identification of isolated bacilli using classical techniques.

Growth Media	Strain	Species
MRS	3	*Lb. paracasei* subsp. *paracasei*
MRS	8	*Lb. brevis*
MRS	9	*Lb. brevis*
MRS	6	*Lb. rhamnosus*
MRS	7	*Lb. rhamnosus*
MRS	11	*Lb. paracasei* subsp. *paracasei*
MRS	10	*Lb. paracasei* subsp. *paracasei*
MRS	12	*Lb. paracasei* subsp. *paracasei*
MRS	13	*Lb. paracasei* subsp. *paracasei*
Μ17	4	*Lb. paracasei* subsp. *paracasei*
Μ17	5	*Lb. paracasei* subsp. *paracasei*
Μ17	6	*Lb. kefiri*
RO	3	*Lb. paracasei* subsp. *paracasei*
RO	4	*Lb. paracasei* subsp. *paracasei*
RO	5	*Lb. paracasei* subsp. *paracasei*
RO	7	*Lb. rhamnosus*
RO	6	*Lb. rhamnosus*
RO	10	*Lb. paracasei* subsp. *paracasei*
RO	11	*Lb. paracasei* subsp. *paracasei*
RO	8	*Lb. rhamnosus*
RO	9	*Lb. rhamnosus*
RO	12	*Lb. paracasei* subsp. *paracasei*
RO	13	*Lb. paracasei* subsp. *paracasei*

**Table 4 foods-12-00370-t004:** Molecular identification of strains.

Raw Milk	(Sugar/Carbohydrate) Fermentation	Multiplex PCR
RO-1	*Leuconostoc pseudomesenteroides*	*Leuconostoc*
RO-2	*Leuconostoc pseudomesenteroides*	*Leuconostoc*
M17-1	*Lactococcus garvieae*	
MRS-1	*Lactococcus garvieae*	*Lactococcus*
MRS-2	*Lactococcus garvieae*	
RO-3	*Lacticaseibacillus paracasei* subsp. *paracasei*	
KAA-1	*Enterococcus faecalis*	*Enterococcus faecalis*
AFTER SALTING		
MRS-3	*Lacto Lacticaseibacillus paracasei* subsp. *paracasei*	*Lactobacillus*
RO-4	*Lacticaseibacillus paracasei* subsp. *paracasei*	
RO-5	*Lacticaseibacillus paracasei* subsp. *paracasei*	
MRS-4	*Lactococcus lactis* subsp. *lactis*	*Lactococcus*
MRS-5	*Lactococcus* spp.	*Lactococcus*
M17-2	*Streptococcus* spp.	*Streptococcus* + *Leuconostoc*
M17-3	*Lactococcus lactis* subsp. *lactis*	*Lactococcus*
KAA-2	*Enterococcus faecalis*	*Enterococcus faecalis*
KAA-3	*Enterococcus gallinarum*	*Enterococcus*
CHEESE (30 DAYS)		
MRS-6	*Lacticaseibacillus rhamnosus*	*Lactobacillus*
MRS-7	*Lacticaseibacillus rhamnosus*	*Lactobacillus*
RO-6	*Lacticaseibacillus rhamnosus*	
M17-4	*Lacticaseibacillus paracasei* subsp. *paracasei*	*Lactobacillus*
M17-5	*Lacticaseibacillus paracasei* subsp. *paracasei*	
RO-7	*Lacticaseibacillus paracasei* subsp. *paracasei*	
MRS-8	*Lacticaseibacillus parabrevis*	*Lactococcus*
MRS-9	*Lacticaseibacillus parabrevis*	*Lactococcus*
M17-6	*Lacticaseibacillus kefiri*	*Leuconostoc*
KAA-4	*Enterococcus faecalis*	*Enterococcus faecalis*
KAA-5	*Enterococcus faecalis*	*Enterococcus faecalis*
KAA-6	*Enterococcus gallinarum*	*Enterococcus*
CHEESE (60 DAYS)		
MRS-10	*Lacticaseibacillus paracasei* subsp. *paracasei*	*Lactobacillus*
RO-8	*Lacticaseibacillus rhamnosus*	*Lactobacillus*
RO-9	*Lacticaseibacillus rhamnosus*	
MRS-11	*Lacticaseibacillus paracasei* subsp. *paracasei*	
RO-10	*Lacticaseibacillus paracasei* subsp. *paracasei*	
RO-11	*Lacticaseibacillus paracasei* subsp. *paracasei*	*Lactobacillus*
M17-7	*Streptococcus* spp.	*Streptococcus* + *Leuconostoc*
KAA-7	*Enterococcus faecium*	*Enterococcus hirae*
CHEESE (90 DAYS)		
MRS-12	*Lacticaseibacillus paracasei* subsp. *paracasei*	*Lactobacillus*
MRS-13	*Lacticaseibacillus paracasei* subsp. *paracasei*	
RO-12	*Lacticaseibacillus paracasei* subsp. *paracasei*	
RO-13	*Lacticaseibacillus paracasei* subsp. *paracasei*	
KAA-8	*Enterococcus faecalis*	*Enterococcus faecalis*
KAA-9	*Enterococcus faecium*	*Enterococcus hirae*

**Table 5 foods-12-00370-t005:** Sequencing results of selected strains after amplification of part of the 16S rRNA gene and comparison with the BLAST database.

Strain	Identification		E Value	Percentage	Accession No
RO-1	*Leuconostoc pseudomesenteroides* strain MC2/2W 16S ribosomal RNA gene, partial sequence	*Leuconostoc pseudomesenteroides*	0.0	99.37%	MF103729.1
MRS-5	*Lactococcus lactis* subsp. *Lactis* KLDS 4.0325 chromosome, complete genome	*Lactococcus lactis* subsp. *Lactis*	0.0	99.37%	CP006766.1
M17-2	*Streptococcus lutetiensis* gene for 16S ribosomal RNA, partial sequence, strain: C57	*Streptococcus lutetiensis*	0.0	99.23%	LC090604.1
Μ17-3	*Lactococcus lactis* strain 4598 16S ribosomal RNA gene, partial sequence	*Lactococcus lactis* subsp. *Lactis*	0.0	96.91%	MT545096.1
MRS-8	*Levilactobacillus parabrevis* strain KM18 16S ribosomal RNA gene, partial sequence	*Levilactobacillus parabrevis*	0.0	99.78%	MN473283.1
MRS-9	*Levilactobacillus parabrevis* strain KM18 16S ribosomal RNA gene, partial sequence	*Levilactobacillus parabrevis*	0.0	98.56%	MN473283.1
M17-6	*Lactobacillus otakiensis* JCM 15,040 gene for 16S ribosomal RNA, partial sequence	*Lentilactobacillus otakiensis*	0.0	99.54%	C480801.1
	*Lactobacillus parabuchneri* strain 2173 16S ribosomal RNA gene, partial sequence	*Lentilactobacillus parabuchneri*	0.0	99.07%	EF535231.1
	*Lentilactobacillus kefiri* strain DH5 chromosome, complete genome	*Lentilactobacillus kefiri*	0.0	98.16%	CP029971.1
MRS-10	*Lacticaseibacillus rhamnosus* strain 14,235 16S ribosomal RNA gene, partial sequence	*Lacticaseibacillus rhamnosus*	0.0	96.15%	MW453828.1
	*Lacticaseibacillus paracasei* subsp. *Paracasei* strain MA51 16S ribosomal RNA gene, partial sequence	*Lacticaseibacillus paracasei* subsp. *Paracasei*	0.0	95.40%	KY425781.1

## Data Availability

Data available upon request to corresponding author.

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
