# Peer review of "Indigenous Lactic Acid Bacteria Isolated from Raw Graviera Cheese and Evaluation of Their Most Important Technological Properties"

_foods, 2023, doi:10.3390/foods12020370_

Round 1

Reviewer 1 Report

The work presented is focussed on bioprospecting novel LAB isolates from milk cheeses. I have some comments which I have mentioned in the manuscript. In addition, it would be great if the safety features of these isolates are also exploited like production of biogenic amines. Further there are no quantification aroma features for which specific aroma tests (like for acetaldehyde and diacetyl) can be conducted. Future application of this study can be explained with respect to type of product formulation, stage of addition, etc. can be discussed a bit further. The comments mentioned within the manuscript.

Author Response

The work presented is focussed on bioprospecting novel LAB isolates from milk cheeses. I have some comments which I have mentioned in the manuscript.

In addition, it would be great if the safety features of these isolates are also exploited like production of biogenic amines.

Reply: This is a great idea. This would be the topic of a future manuscript for the next few months

Further there are no quantification aroma features for which specific aroma tests (like for acetaldehyde and diacetyl) can be conducted.

Reply: The aroma was appreciated by a group of 6 panellists according to reference 19.

Future application of this study can be explained with respect to type of product formulation, stage of addition, etc. can be discussed a bit further. The comments mentioned within the manuscript.

Reply: All your comments have been addressed

How the aroma formation is noted? Is it simply by smelling or using specific tests like that for acetaldehyde or diacetyl?

Reply: The aroma was appreciated by a group of 6 panelists [19].

Mention the incubation temperature at which sugar fermentation was studied?

Reply: the incubation temperature was added

Table-1 does not list the results of proteolysis

Reply: Data not shown was added to the manuscript

Italicize

Reply: Done

Correct spacing

Reply: Done

Why used 650 bp gene specifically? 16s rRNA gene of 1500 bp product could give better coverage and similarity score?

Reply: Primers for the amplification of a larger fragment of the 16s rRNA gene has been tested but we used the ones described by Hou et al., 2018 as more specific for the analyses of 16S rRNA gene from LAB.

Table 5 can go into supplementary

Reply: We consider that it should remain in the manuscript. If you insist it can go to supplementary

Correct the line?

Reply: Done

Further discuss the significance of proteolysis during cheese ripening? Is it desirable by these strains or their is chance of bitter peptides formation

Reply: A phrase was added to the manuscript

Italicize?

Reply: Done

Reviewer 2 Report

The importance and impact of results are not clearly stated! It needs to be improved by specifying the novelty as well as the significance of data and the results.

The clarity of Table presentations is not proper. The scientific names of species is not correctly written in Tables (the genus name and the species name) are stuck together many times!

 Considering the deficiency of novelty and presentation style, the paper does not satisfy the Foods Journal criteria and doesn’t seem suitable for publication in the current form.

Author Response

The importance and impact of results are not clearly stated! It needs to be improved by specifying the novelty as well as the significance of data and the results.

Reply: The manuscript was improved according to the comments of the other reviewers.

The clarity of Table presentations is not proper. The scientific names of species is not correctly written in Tables (the genus name and the species name) are stuck together many times!

Reply: the clarity of the tables and the scientific names of species have been corrected.

Considering the deficiency of novelty and presentation style, the paper does not satisfy the Foods Journal criteria and doesn’t seem suitable for publication in the current form.

Reviewer 3 Report

Line 45. the expression ‘finished’ is not appropriate, it is better to use ‘matured’ or ‘ripened’.

Line 52. The term ‘raw cheese’ should be changed to ‘raw milk cheese’.

Line 54. The abbreviation of ‘lactic acid bacteria’ should be given as LAB and should be used as such in the entire text.

Line 70,73. ‘Cheese’ should be used instead of ‘cheese product’.

Line 144. According to what additional characteristics, how were they evaluated, which references were used.

Line 106. Why was cheese production made 5 times, was 100 bacteria isolated in each cheese production or was a total of 100 bacteria isolated in all of them? please write more clearly.

Line 194. LAB please

Line 194. ‘Colonies’ expression should be used instead of ‘strains’

Line 194-197.  It is poorly expressed, please write it in a better and fluent grammar.

Line 209. How the aroma formation was observed, on what basis the selection was made, please write clearly.

Line 355-365. This part was already mentioned in the introduction section, it seems like there is no need to rewrite it here.

Line 384. Please write lactobacilli instead of lactobacillus

Line 402-407. It is written very simply, a discussion is not included, why the bacterial species obtained are different from reference number 48, for example, it can be discussed.

Line 461. LAB please

Line 463. Raw graviera cheese? Raw milk graviera cheese?

Line 465-468. Please pay attention to the italics of bacteria.

Discussion should be written about the diversity of bacteria isolated at different periods, how did the bacteria isolated during maturation change? Which species was observed more in which period?

When defining, why the difference between phenotypic characterization and characterization by molecular method arises, a short comment should be written about the reason and reference should be made from previous studies that indicate that this situation is normal.

It seems that the spelling of bacteria in table 4 is not italicized, please check the italics in the whole text as well.

Too many paragraphs have been created in short, it looks bad, please combine the paragraphs with each other in a meaningful way.

The explanations under 2.4 are not well written. It is not clear which method was used for whom. It is not correct to give collective references in the first paragraph.

Author Response

Line 45. the expression ‘finished’ is not appropriate, it is better to use ‘matured’ or ‘ripened’.

Reply: Done                         

Line 52. The term ‘raw cheese’ should be changed to ‘raw milk cheese’.

Reply: Done

Line 54. The abbreviation of ‘lactic acid bacteria’ should be given as LAB and should be used as such in the entire text.

Reply: Done

Line 70,73. ‘Cheese’ should be used instead of ‘cheese product’.

Reply: Done

Line 144. According to what additional characteristics, how were they evaluated, which references were used.

Reply: The additional characteristics were curd formation, serum separation and aroma formation according to reference 19

Line 106. Why was cheese production made 5 times, was 100 bacteria isolated in each cheese production or was a total of 100 bacteria isolated in all of them? please write more clearly.

Reply: The cheese production was repeated 5 times according to the experimental design of the study. 100 LAB strains were isolated from all 5 cheese making trials. This is now clearly mentioned in the manuscript

Line 194. LAB please

Reply: Done

Line 194. ‘Colonies’ expression should be used instead of ‘strains’

Reply: Done

Line 194-197.  It is poorly expressed, please write it in a better and fluent grammar.

Reply: The paragraph was rephrased

Line 209. How the aroma formation was observed, on what basis the selection was made, please write clearly.

Reply: The aroma was evaluated by a group of 6 panellists for the appreciation of the aroma.

Line 355-365. This part was already mentioned in the introduction section, it seems like there is no need to rewrite it here.

Reply: Some parts of this paragraph may have already been mentioned but we consider that there is extra information included and it is vital for the discussion.

Line 384. Please write lactobacilli instead of lactobacillus

Reply: Done

Line 402-407. It is written very simply, a discussion is not included, why the bacterial species obtained are different from reference number 48, for example, it can be discussed.

Reply: The paragraph was rephrased

Line 461. LAB please

Reply: Done

Line 463. Raw graviera cheese? Raw milk graviera cheese?

Reply: Done

Line 465-468. Please pay attention to the italics of bacteria.

Reply: Done

Discussion should be written about the diversity of bacteria isolated at different periods, how did the bacteria isolated during maturation change? Which species was observed more in which period?

Reply: This was not included within the scope of the present study. The selection of the LAB strains was based on their technological properties and not the stage of the isolation

When defining, why the difference between phenotypic characterization and characterization by molecular method arises, a short comment should be written about the reason and reference should be made from previous studies that indicate that this situation is normal.

Reply: Differences between molecular and phenotypic characterization have been observed previously in LAB (Fguiri et al., 2015), mainly because of the very high degree of conservation observed in these bacteria and thus the 16S rRNA gene is not sufficient for differentiating between species.

It seems that the spelling of bacteria in table 4 is not italicized, please check the italics in the whole text as well.

Reply: Done

Too many paragraphs have been created in short, it looks bad, please combine the paragraphs with each other in a meaningful way.

Reply: Done

The explanations under 2.4 are not well written. It is not clear which method was used for whom. It is not correct to give collective references in the first paragraph.

Reply: Done

Reviewer 4 Report

Journal                      : Foods

Manuscript ID          : foods-2119748

Manuscript title        : Indigenous Lactic Acid Bacteria isolated from raw Graviera cheese and evaluation of their most important technological properties

In this study, indigenous lactic acid isolates were obtained from raw milk cheese during manufacturing stages and storage period. Then the isolates were characterized for some technological properties and genetically identified. The manuscript is well designed and required discussions have been written properly. The results have importance due to the increasing interest for native starter cultures to be used for fermented dairy products.

However, the potential pathogenicity of the selected starter cultures, especially Enterococcus strains should be more deeply discussed.

Also, due to the new naming of some lactic acid bacterial strains, the names should be checked and corrected. https://isappscience.org/new-names-for-important-probiotic-lactobacillus-species/

On the other hand, determination of the inhibitory effects of selected isolates against various dairy-borne pathogens and spoilage microorganisms will add value to the paper.

Author Response

In this study, indigenous lactic acid isolates were obtained from raw milk cheese during manufacturing stages and storage period. Then the isolates were characterized for some technological properties and genetically identified. The manuscript is well designed and required discussions have been written properly. The results have importance due to the increasing interest for native starter cultures to be used for fermented dairy products.

However, the potential pathogenicity of the selected starter cultures, especially Enterococcus strains should be more deeply discussed.

Reply: Unfortunately, we couldn’t include everything within this manuscript. The pathogenicity of the Enterococcus strains of this study along with extra molecular analysis is the topic of a manuscript that is already under preparation.

Also, due to the new naming of some lactic acid bacterial strains, the names should be checked and corrected. https://isappscience.org/new-names-for-important-probiotic-lactobacillus-species/ 

Reply: The names have been checked and corrected

On the other hand, determination of the inhibitory effects of selected isolates against various dairy-borne pathogens and spoilage microorganisms will add value to the paper.

Reply: This is a great idea but outside of the scope of the present study. We will consider your idea for a future publication

Round 2

Reviewer 2 Report

New version is significantly improved and the authors have revised the manuscript accordingly. It seems fine for publication in the present form.

Author Response

We would like to thank the reviewer for his comments. We are really happy that he approves the changes that we made and that he considers that our manuscript is now ready for publication

Reviewer 4 Report

I did not mean adding new data about the pathogenecity of the isolates. The authors must give some discussion about this issue.

Author Response

We would like to thank the reviewer for his comments. The aim of the present study was to isolate, identify and characterize LAB isolates from raw-milk starter-free cheeses with typical sensorial profiles, to evaluate some of their technological properties with those of commercial cultures and finally to select a few ‘wild’ LAB strains that could potentially be used as starter cultures. We appreciate your comments, but we consider that the data included within the manuscript about the pathogenicity of the isolates cannot justify the extension of its discussion further. 

As we have already mentioned, the pathogenicity of the Enterococcus strains of this study along with extra molecular analysis is the topic of a manuscript that is already under preparation. Since we are probably going to publish it in a MDPI journal, a good idea would be to send it to you for revision! We would be more than happy if you could review it as well!

Thank you again and we sincerely hope that our answer won't disappoint you.